# Multicriteria Assessment of the Quality of Waste Sorting Centers—A Case Study

**Karmen Pažek** [1,*]**, Jernej Prišenk** [1]**, Simon Bukovski** [2]**, Boris Prevolšek** [3] **and Črtomir Rozman** [1]

[1] Department of Agricultural Economics and Rural Development, Faculty of Agriculture and Life Sciences, University of Maribor, Pivola 10, 2311 Hoče, Slovenia; jernej.prisenk@um.si (J.P.); crt.rozman@um.si (Č.R.)

[2] Environmental Protection, Ecology Department, KOVA d.o.o., 3000 Celje, Slovenia; simon.bukovski@gmail.com

[3] Faculty of Tourism, University of Maribor, Cesta Prvih Borcev 36, 8250 Brežice, Slovenia; boris.prevolsek@um.si

* Correspondence: karmen.pazek@um.si

**Abstract:** In this paper, the quality of the municipal waste sorting process in seven waste management centers in Slovenia was assessed using the qualitative multicriteria analysis (MCA) method DEX (Decision EXpert) implemented in DEXi software, which is based on multicriteria decomposition of the problem and utility functions in the form of "if–then" decision rules. The study was based on eight types of secondary raw materials. The quality of the secondary raw materials, the regularity of the delivery of secondary raw materials to recycling units based on the sorting efficiency, and the loading weight of the individual baled fractions in the transport of secondary raw materials for recycling were the main parameters used in the model. The final assessment shows "good" waste management service in centers A and D. Centers B, C, and F were rated "average". The "bad" rating was assigned to centers E and G.

**Keywords:** waste sorting; quality management; multicriteria decision analysis; expert system DEXi

## 1. Introduction

The quantity of solid waste is increasing and its management has become a serious issue in modern industrial society. Furthermore, the extension of existing facilities may not be always possible. Therefore, there is a clear need for a reaction from authorities, with the implementation of waste generation reduction, recycling programs, and new facilities. The introduction of new facilities is related to conflicts, and the decision is usually met with considerable local opposition. The issue becomes even more complex due to the lack of appropriate, qualitative and quantitative, environmental assessment tools [1].

The environmental impact of each production phase is an important problem in the agro industrial sector (the use of nonrenewable resources and their impact must be assessed in order to ensure technical sustainability and the economic feasibility of the proposed solutions). The evaluation is mostly conducted with life cycle assessment (LCA methodology), which is one of the most common and efficient tools for environmental analysis [2].

According to Eurostat data from [3] 2016 in the European Union, one individual generates 475 kg of municipal waste (2014 data), and 44% of that waste is processed (recycling and composting).

The authors of [4] provided a scenario analysis using a multiperiod waste management multiobjective optimization, considering economic and environmental issues. They evaluated the type of energy provision, the location of recycling facilities, and the application of recycled material confirmed the ranking of results with respect to global warming potential and total costs. Through

sensitivity analysis with respect to input data, it was revealed that nine parameters were typically sufficient to achieve more than 90% of the total variance of the results. Sorting efficiency, technical yields, and market substitution factors were the most critical parameters. Environmental and financial benefits are possible when a high quality of recycled plastic is achieved.

A life cycle assessment was used to evaluate the possible environmental impacts of the existing and planned plastic waste management scenarios on various impact categories for the study area, the city of Dhanbad in India [5]. Two major plastic waste products were observed, polyethylene terephthalate (PET) and polyethylene (PE). Appropriate recycling was assessed with respect to the environmental impact on most of the impact criteria, due to the use of recycled PET and PE flakes as substitutions for virgin PET and PE flakes. The application of predictive and prognosis models is useful, providing reliable support for decision-making processes. Some indicators such as the number of residents, population age, urban life expectancy, and total municipal solid waste were used as input variables in prognostic models in order to predict the amount of solid waste fractions, as presented in [6], the authors of which used regression analysis and time series analysis to forecast municipal solid waste generation (waste prognostic tools). Regression equations were determined for six solid waste fractions (paper, plastic, metal, glass, biodegradable material, and other waste).

The development of waste management systems is often related to multiple conflicting criteria and to multiple decision-makers. Moreover, due to the rapid and intensive development of existing and new technologies with respect to waste management problems, decision-makers must select from a wide spectrum of available alternatives [7]. Therefore, multicriteria modeling is a key element of decision support, providing a formal structure of existing knowledge or problems related to impacts, identifying gaps, rankings, etc. [8–10]. In this light, the authors of [11] proposed a selection of proper waste management systems using the multicriteria analysis (MCA) method ELECTRE. MCA is useful when we have to deal with multiple conflicting parameters and multiple decision-makers. MCA has been developed rapidly over the last 30 years, and it is able to consider several consequences of proposed solutions of various typologies of problems [2].

For efficient solid waste management, a detailed screening of needs and desired development directions followed by implementation decisions are required. As a result of this process, various solid waste management scenarios have been proposed, and many of them may have mutually conflicting objectives. However, multiattribute decision-making methodologies and models have become convenient tools supporting solid waste management because they can handle problems involving multiple dimensions and conflicting criteria, as reported by [4,7,12–21] and others.

Scenarios affecting different population groups lead to diverse problems, varying in the costs and time required to become effective. Usually, different groups of decision-makers are involved for scenario selection. Decision-making usually has to take into account conflicting technological, economic, social, and environmental criteria. Single-criterion decision-making that is based only on the available financial resources as a sole criterion cannot respond to such requirements. In [22], the authors demonstrated the possibilities of using a multicriteria decision-making tool to select the best municipal solid waste management scenario among six different alternatives. This multicriteria decision-making tool in this study enables decision-makers to make informed decisions and achieve optimal results. Further, EUGÈNE, a sophisticated mixed integer linear programming model, was developed in [23]. The aim of this model is to support regional decision-makers in long-term planning for solid waste management activities. The model removes almost every limitation encountered in other waste management models and contains a large quantity of variables and constraints. The method used to embed waste management environmental parameters in the EUGÈNE model consists in building a global impact index (GII) for all site/facility combinations. First, an environmental and spatial evaluation of waste management facilities over sites is based on qualitative and quantitative criteria measuring biophysical and social impacts. Spatial analysis is carried out by geographical information system routines. A MCA ranks all site/facility combinations according to their global performance based on all criteria. The net flow, computed by the PROMETHEE multicriteria outranking method, is

considered as a GII to be embedded into EUGÈNE. The model objective function is thus modified to minimize total system cost and GII.

Environmental issues, such as waste management, when uncertainty is involved also tend to be a suitable field for the application of multicriteria decision analysis techniques. The authors of [24] presented two cases where decision-makers had different preferences. Social agents required an assessment of plastic waste disposal alternatives, while a performance evaluation for existing construction and demolition facilities was required in the second case. The analysis was based on multicriteria evaluation with the support tool THOR. THOR ranks the alternatives from best to worst for each criterion considered in the analysis. The THOR system is based on the aggregation of the multiattribute utility theory (MAUT) and preference modeling.

A model for the assessment of the sustainability of waste treatment scenarios with the use of an analytical hierarchical process (AHP) is described in [8]. The model predicts an increase in the number of indicators and a new criterion for the selection of indicators: the relevance of the indicator for a certain waste treatment. The composting and recycling of inorganic waste were assessed as the best scenario. The AHP for waste management assessment was also used in [25–28]. A complete literature review of using MCA for waste management is provided in [7], the AHP being the most widely applied MCA method.

A decision-making model should enable the evaluation of all options when taking into account all factors that influence decisions. A multicriteria decision analysis (MCDA) can be used to assess decisions related to organization and planning. Most of the methods result in quantitative assessment, while the DEX method developed in [29] uses qualitative criteria values and a utility function in the form of "if–then" decision rules. In this light, the benefits of strategic environmental considerations in the process of siting a repository for low- and intermediate-level radioactive waste (LILW) are presented in [14]. The site selection processes were compared with the support of the decision expert system DEX. Further application of MCDA in environmental management has been discussed and applied in complex real-word environmental decision problems and presented in the literature. We can observe that the AHP and Electre have been used in addition to DEXi in tackling decision support in the area of waste management [9,30–33].

The hierarchical decision model for quality in municipal waste sorting service assessment was developed in [34], which was the author's master thesis for the Faculty of Agriculture and Life Sciences, University of Maribor.

The aim of this paper is a multicriteria assessment of the municipal waste sorting service in Slovenia using the qualitative DEX method. In Slovenia, the basic separation is conducted at households: glass, plastic bottles, and so on, organic material, and the remaining general waste, which is then sorted in centers. We examine eight types of secondary raw materials. The assessment parameters were the quality of the secondary raw materials determined by their purity, the regularity of the delivery of secondary raw materials to recycling units, based on the sorting efficiency, and the loading weight of the individual baled fractions in the transport of secondary raw materials for recycling, which fundamentally depends on the technical equipment of the centers. This paper is organized as follows: in the first part, the article describes the development and application of the DEX-based multicriteria model. The model development is presented in the methodology section. The MCA of seven centers (A to G) using the DEX model is presented in the results section of the article, while main findings and final remarks conclude the article.

## 2. Materials and Methods

### 2.1. Case Study Situation

In the past, most of the waste was landfilled in Slovenia. Today, certain landfills are already closed or are in the process of closure. Eurostat states that the largest share of waste was in the EU in 2002, when each European citizen created 527 kg of municipal waste annually. Statistics have shown that,

since 2007, the quantity of municipal waste per capita decreases every year. According to data for 2014, Slovenia, with 432 kg (or 1.3 kg/day/inhabitant) of municipal waste produced annually, ranks slightly below the European Union average (475 kg/year), as presented in Table 1. The lowest value per quantity of recycled waste per Slovenian inhabitant was achieved in 2001, that is, 7 kg/inhabitant/year, and the highest value was found in 2015 (208 kg/inhabitant/year).

**Table 1.** Production of municipal waste per inhabitant of Member States of the European Union [3].

| | Generated. (kg per Person) | Treated, (kg per Person) | Municipal Waste Treated, % | | | |
|---|---|---|---|---|---|---|
| | | | Recycled (%) | Composted (%) | Incinerated (%) | Landfilled (%) |
| **EU** | **475** | **465** | **28** | **16** | **27** | **28** |
| Belgium | 435 | 439 | 34 | 21 | 44 | 1 |
| Bulgaria | 442 | 416 | 23 | 2 | 2 | 74 |
| Czech Republic | 310 | 310 | 23 | 3 | 19 | 56 |
| Denmark | 759 | 759 | 27 | 17 | 54 | 1 |
| Germany | 618 | 618 | 47 | 17 | 35 | 1 |
| Estonia | 357 | 303 | 31 | 6 | 56 | 8 |
| Ireland | 586 | 531 | 34 | 6 | 18 | 42 |
| Greece | 509 | 509 | 16 | 4 | 0 | 81 |
| Spain | 435 | 435 | 16 | 17 | 12 | 55 |
| France | 511 | 511 | 22 | 17 | 35 | 26 |
| Croatia | 387 | 374 | 15 | 2 | 0 | 83 |
| Italy | 488 | 455 | 28 | 18 | 21 | 34 |
| Cyprus | 626 | 626 | 13 | 12 | 0 | 75 |
| Latvia | 281 | 281 | 3 | 5 | 0 | 92 |
| Lithuania | 433 | 425 | 21 | 10 | 9 | 60 |
| Luxemburg | 616 | 616 | 28 | 18 | 35 | 18 |
| Hungary | 385 | 376 | 25 | 6 | 10 | 59 |
| Malta | 600 | 545 | 8 | 4 | 0 | 88 |
| Netherlands | 527 | 527 | 24 | 27 | 48 | 1 |
| Austria | 565 | 547 | 26 | 32 | 38 | 4 |
| Poland | 272 | 272 | 21 | 11 | 15 | 53 |
| Portugal | 453 | 453 | 16 | 14 | 21 | 49 |
| Romania | 254 | 214 | 5 | 11 | 2 | 82 |
| Slovenia | 432 | 257 | 49 | 12 | 0 | 39 |
| Slovakia | 321 | 282 | 6 | 6 | 12 | 76 |
| Finland | 482 | 482 | 18 | 15 | 50 | 17 |
| Sweden | 438 | 438 | 33 | 16 | 50 | 1 |
| United Kingdom | 482 | 473 | 28 | 17 | 27 | 28 |
| Iceland | 345 | 345 | 38 | 7 | 6 | 49 |
| Norway | 423 | 414 | 27 | 17 | 54 | 3 |
| Switzerland | 730 | 730 | 33 | 21 | 46 | 0 |
| Montenegro | 508 | 451 | 1 | 0 | 0 | 99 |
| F Y R of Macedonia | 370 | 370 | | | | 100 |
| Serbia | 302 | 236 | 1 | 0 | 0 | 99 |
| Turkey | 405 | 363 | 0 | 0 | 0 | 100 |
| Bosnia & Herzegovina | 349 | 234 | 0 | 0 | 0 | 100 |

Municipal waste from households is taken over for processing by the waste management centers. There are quite a few modern centers in Slovenia that were either modernized or newly built in the last few years. Waste management centers are the first to take care of the waste after the consumer's use. In Slovenia, 12 centers for waste management were planned. For eight centers, the implementation was planned through cohesion projects, and four centers should be financed from the budgets of the municipalities involved. An appropriate and thus desirable methodology for the assessment of the quality of the process combines the available data from waste centers with multicriteria decisions about holistic waste sorting management.

Municipal waste management centers accept solid municipal waste in four groups: mixed municipal waste, separately collected biodegradable waste, bulky waste, and separately collected fractions. Separately collected fractions are further divided into two groups, namely waste packaging and waste electrical and electronic equipment. Regardless of the separate collection already at the source, the separately collected fraction is sorted before being handed over for recycling to remove any impurities and further separate the collected packaging waste by material type and into other subgroups. After sorting, we obtain waste that is suitable for recycling, waste that is intended for energy use, and some waste for disposal.

*2.2. Model Development*

Environmental decisions demand multicriteria assessment models that are based on quantitative data or qualitative judgments of values. Multicriteria decision-making requires establishing preferences and priorities and selecting from the set of available alternatives with respect to many criteria that are usually conflicting [24]. DEX is an expert system shell for qualitative multiattribute decision modeling and support [35]. In [29], it was shown that the DEX methodology is appropriate for soft, that is, less structured and less formalized, decision problems.

The main feature of DEX (Decision EXpert) is the use of scales with qualitative values for all defined criteria, for instance, "low", "average", and "high". without numerical scales. Usually the scales are ordinal; however, nominal scales can also be used. The DEX methodology is implemented in the program called DEXi (DEX for Instruction) [36,37]. In the DEXi model, the attributes are organized hierarchically. The attributes are assigned with qualitative scales, and utility functions (aggregation functions) are defined in the form of "if–then" decision rules (decision tables). The decision rules are defined for each combination of input attributes. When the alternatives are evaluated, attribute values are put in at the lowest level of the hierarchy. Aggregation functions calculate the aggregated attribute values from bottom-up through the entire hierarchy [38].

DEXi is user-friendly and enables convenient model construction in cooperation between the decision analyst and experts from the field. The expert defines attributes and decision rules, while the analyst coordinates the process and constructs the model. The model is developed in four steps [7]:

(1)   identification of attributes,
(2)   building the hierarchy (Figure 1),
(3)   defining scales for each attribute scales (Figure 2), and
(4)   defining utility functions (decision rules) (Figures 3 and 4).

The model for material selection consists of 43 hierarchical structured attributes organized into a hierarchical tree (Figure 2). The basis of the model is represented by four basic attributes: PET, low-density polyethylene (LDPE) + high-density polyethylene (HDPE), polystyrene (PS) and polypropylene (PP) PP/PS + hard mixed plastics (toys, plastic cases, etc.), and other materials. These attributes are further divided into additional sub-attributes, and those are additionally divided into leaves in the hierarchical tree.

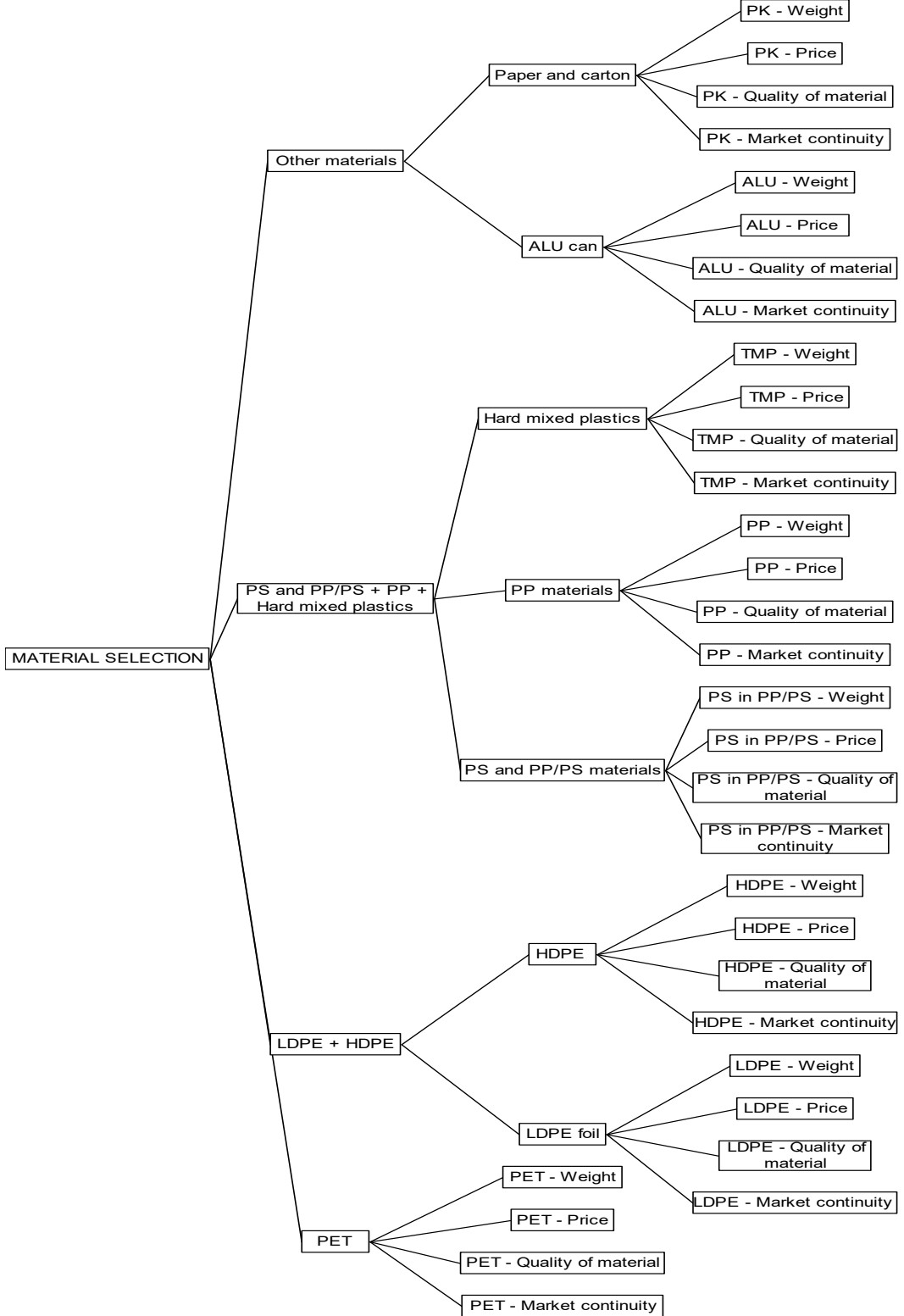

**Figure 1.** Hierarchical structure of attributes tree for the assessed problem.

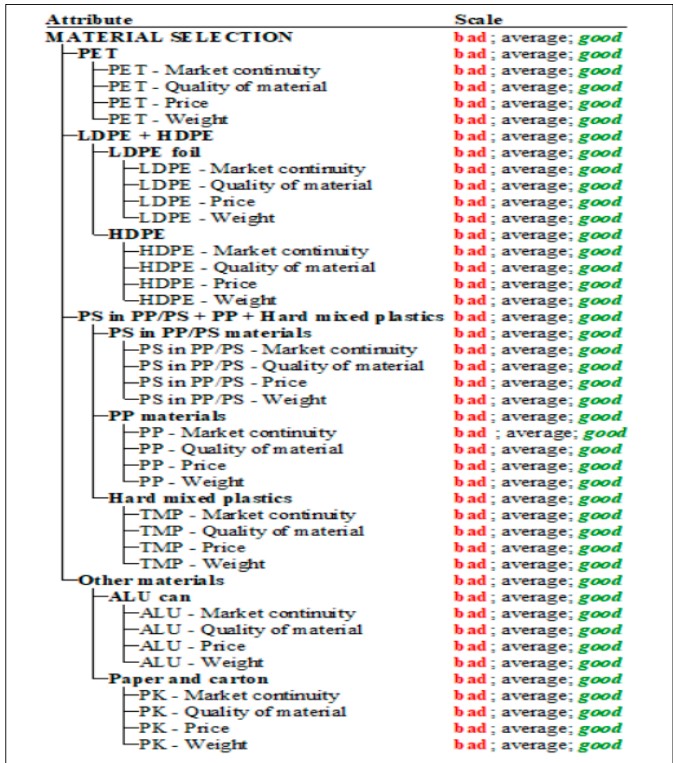

**Figure 2.** Definition of attribute value scales.

| | PET | LDPE + HDPE | PS in PP/PS + PP + Hard mixed plastics | Other materials | MATERIAL SELECTION |
|---|---|---|---|---|---|
| | 31% | 31% | 15% | 23% | |
| 1 | bad | bad | bad | * | bad |
| 2 | bad | bad | <=average | <=average | bad |
| 3 | bad | bad | * | bad | bad |
| 4 | bad | <=average | <=average | bad | bad |
| 5 | <=average | bad | <=average | bad | bad |
| 6 | bad | * | >=average | good | average |
| 7 | <=average | <=average | >=average | good | average |
| 8 | * | bad | >=average | good | average |
| 9 | bad | * | good | >=average | average |
| 10 | <=average | <=average | good | >=average | average |
| 11 | <=average | * | good | average | average |
| 12 | * | bad | good | >=average | average |
| 13 | * | <=average | good | average | average |
| 14 | bad | >=average | * | >=average | average |
| 15 | <=average | average | * | >=average | average |
| 16 | <=average | >=average | bad | >=average | average |
| 17 | <=average | >=average | * | average | average |
| 18 | * | average | bad | >=average | average |
| 19 | * | average | * | average | average |
| 20 | * | >=average | bad | average | average |
| 21 | bad | >=average | good | * | average |
| 22 | <=average | average | good | * | average |
| 23 | <=average | >=average | good | <=average | average |
| 24 | * | average | good | <=average | average |
| 25 | bad | good | * | * | average |
| 26 | <=average | good | bad | * | average |
| 27 | <=average | good | * | <=average | average |
| 28 | * | good | bad | <=average | average |
| 29 | * | good | <=average | bad | average |
| 30 | average | <=average | * | >=average | average |
| 31 | average | * | bad | >=average | average |
| 32 | average | * | * | average | average |
| 33 | >=average | bad | * | >=average | average |
| 34 | >=average | <=average | bad | >=average | average |
| 35 | >=average | <=average | * | average | average |
| 36 | >=average | * | bad | average | average |
| 37 | average | <=average | good | * | average |
| 38 | average | * | good | <=average | average |
| 39 | >=average | bad | good | * | average |
| 40 | >=average | <=average | good | <=average | average |
| 41 | average | average | * | * | average |

**Figure 3.** Decision rules that determine the main goal.

**Decision rules**

| PET - Market continuity 30% | PET - Quality of material 20% | PET - Price 20% | PET - Weight 30% | PET |
|---|---|---|---|---|
| 1 bad | bad | <=average | <=average | bad |
| 2 bad | bad | * | bad | bad |
| 3 bad | <=average | bad | <=average | bad |
| 4 bad | <=average | <=average | bad | bad |
| 5 bad | * | bad | bad | bad |
| 6 <=average | bad | <=average | bad | bad |
| 7 <=average | <=average | bad | bad | bad |
| 8 bad | * | * | good | average |
| 9 <=average | bad | * | good | average |
| 10 <=average | <=average | <=average | good | average |
| 11 <=average | * | bad | good | average |
| 12 * | bad | <=average | good | average |
| 13 * | <=average | bad | good | average |
| 14 bad | * | good | >=average | average |
| 15 <=average | bad | good | >=average | average |
| 16 <=average | * | good | average | average |
| 17 * | bad | good | average | average |
| 18 bad | >=average | >=average | >=average | average |
| 19 <=average | average | average | >=average | average |
| 20 <=average | >=average | >=average | average | average |
| 21 * | average | average | average | average |
| 22 bad | >=average | good | * | average |
| 23 <=average | >=average | good | <=average | average |
| 24 * | >=average | good | bad | average |
| 25 bad | good | * | >=average | average |
| 26 <=average | good | bad | >=average | average |
| 27 <=average | good | * | average | average |
| 28 * | good | bad | average | average |
| 29 bad | good | >=average | * | average |
| 30 <=average | good | >=average | <=average | average |
| 31 * | good | >=average | bad | average |
| 32 average | bad | * | >=average | average |
| 33 average | <=average | <=average | >=average | average |
| 34 average | * | bad | >=average | average |
| 35 average | * | * | average | average |
| 36 >=average | bad | <=average | >=average | average |
| 37 >=average | bad | * | average | average |
| 38 >=average | <=average | bad | >=average | average |
| 39 >=average | <=average | <=average | average | average |
| 40 >=average | * | bad | average | average |
| 41 average | bad | good | * | average |
| 42 average | * | good | <=average | average |

**Figure 4.** Decision rules that determine the PET category.

Plastic and other materials (Al, paper, cartons, etc.) have become an important issue for local and government environmental policy. This material is not entirely separated at the source (households). Reuse and recycling have been seen as the best options for effectively solving the issue of waste in sorting centers. However, the hierarchical structure of the attributes tree for the observed problem will enable an assessment of the quality of the sorting of municipal waste in seven waste management centers.

The assessment was based on eight types of secondary raw materials (PET; LDPE foil, a thermoplastic made from the monomer ethylene; HDPE, a thermoplastic polymer produced from the monomer ethylene; polystyrene (PS) in PP/PS materials, PP material; hard mixed plastics; Al cans; and paper and cartons). The assessment parameters were as follows: quality of the secondary raw materials determined by their purity, the regularity of the delivery of secondary raw materials to recycling units, based on the sorting efficiency, and the loading weight of the individual baled fractions in the transport of secondary raw materials for recycling, which fundamentally depends on the technical equipment of the centers.

Since the DEXi operates with discrete values, the classification must be performed. The process of classification determines the qualitative value according to the defined list of values for each attribute. The classification could be numerical or non-numerical (qualitative). In the presented case, all value scales are defined according to the waste center's database and consist of data from their practices. The categorization for scale values is defined for the lowest level of the hierarchy, and the values of aggregate attributes are determined by utility functions.

### 2.3. Categorization of Input Attributes

The value scales by market continuity are defined according to the annual accessibility of material, as follows: bad: rarely in the market, and seasonal or sufficient stock for dispatch only after more than two months; average: occasionally on the market, and sufficient stock for dispatch from more than one month to a maximum of two months; good: regularly on the market, and ready for shipping each month or several times a month.

The quality of secondary raw material depends on the purity of the individual fraction, which is basically influenced by the quality of the input material and later by the sorting quality. The more monotonous waste is, the higher the quality of the material (i.e., bad: the quality of the material is poor, with many of admixtures; average: the quality of the material is average, with observed impurities; good: the quality of the material is good and free of impurities). The price of a particular fraction depends on how "clean" the sorted material is. The prices of fractions are most influenced by the quality of the raw materials, and in the analyzed case, we used a value scale with bad, average, or good). The maximum price (Pmax) was the basis for categorization of the price attribute: prices lower than one-third of Pmax were classified as bad, prices between one-third of Pmax and two-thirds of Pmax were classified as average, and prices higher than two-thirds of Pmax were classified as good.

At last, the loading weight was determined by the three-value scale (bad, average, or good). The higher the loading weight is, the lower the cost of transport per tons of waste is, and the higher the offered price for the purchase is. The loading weight of secondary raw materials is defined by the following value scale: bad: loading weight < 13 t; average: loading weight between 13 and 17 t; good: good loading weight > 17 t.

The utility function for the whole model (material selection) is composed of many partial utility functions that are defined for all aggregate attributes (defined by "if–then" decision rules). For the quality of the municipal waste sorting service problem, a series of 81 decision rules was identified, estimating the overall project evaluation for each possible value combination of aggregate attributes (Figure 4). The rules are presented in aggregate form, where * means any value, >= means equal or better, and <= means equal or worse. The utility function is defined through the entire hierarchy for each aggregate attribute. The decision rule describes the value of an aggregate attribute for each combination of input attributes and expresses the relative importance of individual attributes. For a less detailed representation of utility functions, the weights can be used. Given a decision rule (such as that in Figure 4), we used a suitable method to estimate the average importance of each input attribute for determining the value of a dependent variable. We then obtained weights by expressing this importance as percentages relative to each of the other attributes. Two methods were used to assess weights with DEX: one was based on regression and the other on measuring attribute informativity as in machine learning methods [36]. The rules were set up together by model developers and experts from each center after multiple discussions in focus groups. General agreement was reached on most of the decision rules, representing the knowledge of experts and their preferences with respect to the assessment problem. However, the experts were not asked to argue for their preferences.

The attributes at the lowest level describe the alternatives at the basic level. They represent input data for the model and are assessed by the decision-maker. The heading in Figure 4 that determines the material selection goal shows the approximate attribute weights derived from the decision rules. The * defined any value, >= means better or equal, and <= means worse or equal. In the expert system DEXi, "bad" values are in red, "average" or "middle" values are in black, and "good" values are usually in green and italic. Decisions about the weights for main attributes significantly influenced the results (the ranking of quality of municipal waste sorting services).

As mentioned before, there were 81 decision rules for the assessment of material selection determined by an expert group in waste centers. After decision rules were established, the decision-maker put in qualitative values for each attribute corresponding to each decision scenario. Once the values were inserted, the DEXi performed the analysis for each decision alternative (in

the presented case assessment). As can be seen in Figure 3, the importance of attributes can also be represented by weights.

According to the defined decision rules and their representation by weights, the PET criteria have equal importance as LDPE + HDPE (31%), followed by the importance of other material and the last PS + PP/PS + hard mixed plastics with 15%. In the next figure (Figure 4), the decision rules for only one category are presented in more detail.

The aggregation of values was conducted according to decision rules, which are usually presented in the form of a decision table. An example for the observed case study is presented in Figures 3 and 4.

As can be seen in the last figure, market continuity and weight criteria are the most important (30%), and a similar weight was determined by quality material and price criteria (20%). The rules were determined by the expert group.

*2.4. Data Sources*

The authors observed and analyzed seven waste sorting centers in Slovenia. The input data were obtained by center management and its databases. By the definition of the attributes, several communication and meetings were provided by the expert group. In the first phase of model attribute definitions, many changes in the model structure and aggregation rules were made.

Based on separately collected fractions from secondary raw materials, seven waste management centers in Slovenia (A–G) were evaluated.

Assessment criteria are as follows:

- the quality of municipal waste sorting after sorting extracted secondary raw materials,
- the regularity of the material on the market or the quantity of materials intended for processing, and
- the loading weight of baled secondary raw materials.

## 3. Results and Discussion

In this research, more emphasis is placed on the third point in the five-level scale of waste management, that is, the recycling process. Collected secondary raw materials are quite different in the waste management centers, regardless of the fairly similar requirements between processors. Differences from the mixed municipal waste in the sorting method, the purity of the fractions, the percentage of the obtained fractions, and the percentage of secondary raw materials are apparent.

Waste from the waste management centers is purchased by different waste collectors, dealers, and brokers, which ensure that waste separated by fractions ends up in recycling processes throughout Europe. However, processors have certain requirements for taking up waste. If the quality of the waste meets the requirements, the waste is redeemed and processed; otherwise, the waste shipment may also be rejected.

Most of the waste comes from households. Municipal waste is waste from the household or similar waste from production, trade, services, or other activities by nature or composition. Municipal waste could be said to be waste generated solely by the needs of people in a household, from household and non-household activities. However, municipal waste cannot be classified as waste from a production or service process [34].

Waste management systems (according to legislation [39]) are the same in all centers across Slovenia, but in the provision of waste management services, there are differences in the quality of secondary raw materials.

We observed and analyzed seven waste sorting centers in Slovenia. Some basic information about the centers is provided in Table 2.

**Table 2.** Basic information on waste management centers.

| Waste Center | Process |
|---|---|
| A | <ul><li>Reception of mixed municipal waste</li><li>Disposal of mixed municipal waste</li><li>Waste collection and treatment</li><li>Sorting waste by fractions</li><li>Open and closed composting</li><li>Mechanical and biological treatment of mixed municipal waste</li><li>Baling of waste after sorting</li><li>Burning of sorted waste</li></ul> |
| B | <ul><li>Reception of mixed municipal waste</li><li>Sorting mixed municipal waste</li><li>Composting</li><li>Disposal of residues from sorting into landfill</li></ul> |
| C | <ul><li>Reception of mixed municipal waste</li><li>Sorting mixed municipal waste</li><li>Composting</li><li>Disposal of residues from sorting into landfill</li></ul> |
| D | <ul><li>Collection of municipal waste</li><li>Collection of mixed municipal waste</li><li>Collection of municipal waste from activities (crafts, industry, and public institutions)</li><li>Collection of mixed waste packaging</li><li>Sorting waste</li><li>Mechanical and biological treatment of waste</li><li>Separation of heavy fractions into a landfill</li><li>Waste balancing is sorting</li><li>Burning of sorted waste</li><li>Brokering of secondary raw materials for processing</li></ul> |
| E | <ul><li>Sorting mixed municipal waste</li><li>Separate collection of fractions and bulky waste</li><li>Storage of baled secondary raw materials</li><li>Composting</li><li>Storage of ready-made compost</li><li>Processing of inert construction waste</li></ul> |
| F | <ul><li>Collection of mixed municipal waste</li><li>Sorting separately collected waste fractions</li><li>Waste compression</li><li>Bio-waste composting</li><li>Mechanical and biological waste treatment (heavy and light fractions)</li><li>Transfer of light fractions of mixed municipal waste for thermal treatment to another company</li><li>Transfer of heavy fractions of mixed municipal waste to landfill</li><li>Storage of secondary raw materials, bulky waste, and hazardous substances</li></ul> |
| G | <ul><li>Disposal and collection of mixed municipal waste</li><li>Sorting mixed municipal waste</li><li>Bio-waste composting</li><li>Compost storage</li><li>Storage of baled secondary raw materials</li><li>Waste incineration</li><li>Waste recycling</li><li>Waste disposal at a regional center</li></ul> |

Other materials that are processed in those centers are as follows:

- paper, cardboard, and paper packaging of all types and sizes,
- glass and glass packaging of all types and sizes,
- plastic and plastic packaging of all types and sizes,
- metal waste and metal packaging of all types and sizes (non-ferrous metals),
- waste film, also contaminated,
- Styrofoam,
- wood and wood waste and wooden packaging of all types and sizes,
- clothing and textiles,
- garden cuts,
- reduced amounts of construction waste,
- all kinds of tires, used rims, and other waste,
- waste electrical and electronic equipment,
- bulky waste (furniture and other household waste)

The input data were obtained by the center managements and their databases.

A final assessment of the seven options is presented in Figure 5.

| Option | A | B | C | D | E | F | G |
|---|---|---|---|---|---|---|---|
| PET - Market continuity | good | average | good | good | bad | good | average |
| PET - Quality of material | good | average | good | good | bad | average | bad |
| PET - Price | average | bad | average | bad | average | bad | average |
| PET - Weight | average | average | bad | good | average | bad | bad |
| LDPE - Market continuity | good | average | * | good | bad | good | good |
| LDPE - Quality of material | good | bad | * | good | bad | bad | bad |
| LDPE - Price | good | average | * | bad | average | bad | bad |
| LDPE - Weight | good | good | * | good | good | good | good |
| HDPE - Market continuity | good | average | good | good | bad | average | average |
| HDPE - Quality of material | good | average | good | average | bad | bad | bad |
| HDPE - Price | good | average | bad | average | bad | average | average |
| HDPE - Weight | average | bad | bad | good | bad | bad | bad |
| PS in PP/PS - Market continuity | good | bad | good | good | bad | average | average |
| PS in PP/PS - Quality of materia | good | bad | good | good | bad | average | bad |
| PS in PP/PS - Price | good | average | average | bad | average | average | average |
| PS in PP/PS - Weight | bad | bad | bad | average | bad | bad | bad |
| PP - Market continuity | average | average | average | good | * | average | average |
| PP - Quality of material | good | good | good | good | * | bad | bad |
| PP - Price | good | average | good | average | * | average | average |
| PP - Weight | bad | bad | average | good | * | bad | bad |
| TMP - Market continuity | good | average | good | * | * | * | bad |
| TMP - Quality of material | good | good | good | * | * | * | bad |
| TMP - Price | good | good | bad | * | * | * | good |
| TMP - Weight | bad | average | bad | * | * | * | bad |
| ALU - Market continuity | good | good | good | good | bad | good | good |
| ALU - Quality of material | good | good | good | good | bad | bad | bad |
| ALU - Price | bad | bad | good | good | bad | average | bad |
| ALU - Weight | average | average | average | average | average | average | average |
| PK - Market continuity | * | * | good | * | bad | good | average |
| PK - Quality of material | * | * | good | * | bad | average | average |
| PK - Price | * | * | average | * | average | average | bad |
| PK - Weight | * | * | good | * | good | good | good |

**Figure 5.** Assessment options for waste sorting ranking of waste centers (**A**–**G**).

As presented in Figure 6, the final assessment showed that differences in the quality of the waste management service between the centers exist. The results of the DEXi model show that waste centers A and D were assessed as "good". Waste center C was the highest center assessed as "average", followed by waste centers B and F. Waste centers E and G were assessed as "bad".

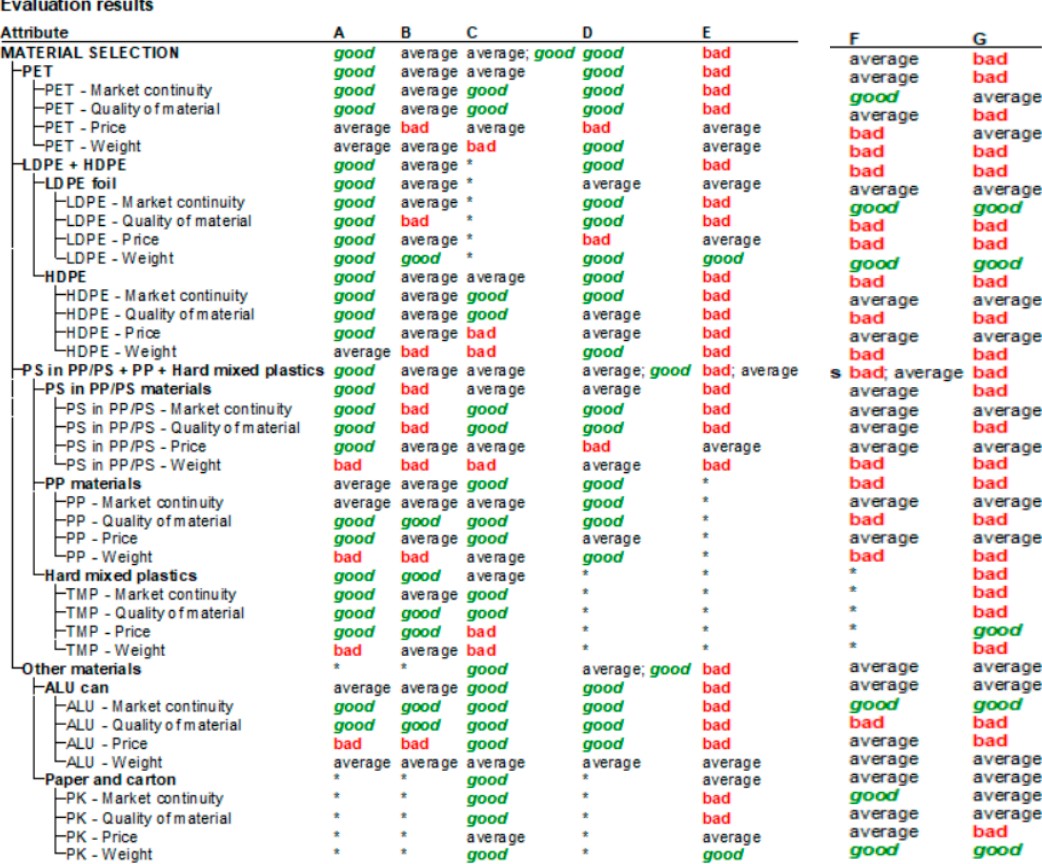

**Figure 6.** Final assessment of waste sorting service in Slovenia.

The quality of secondary raw materials was assessed on the basis of how clean the fractions obtained by the centers after sorting were. The content of the admixture, whether they followed the specifications of the processors, and whether they produced several types of baled materials for processing, because the combination of various materials also reduced their quality. The second parameter of the assessment was the regularity of the material on the market, which basically depends on the quantity of the input material that the centers take over. However, the efficiency of waste sorting is related to the quantity of secondary raw materials. The third parameter was the assessment of the loading weight of the secondary raw materials. Since most of the waste is delivered to processors outside Slovenia, the loading weight in the transport is essential because of the economic efficiency of waste recycling. We can observe that centers A and D received the highest score. Both of those centers also burn sorted waste. However, it is interesting to observe that center D is from the central region, which has the highest GDP per capita, and that center A is in the east in one of the most underdeveloped regions. This is in contrast with the expectation that households with a higher income produce more waste considered "quality." This can be explained by the fact that the model is more oriented towards an assessment of the process and was not aimed at efficiency in the form of an input/output ratio. Various technical equipment of the waste management centers is influenced by the different loading weight of individual secondary raw materials. The results of the DEXi model show that, despite the requirements of the Waste Directive, there are noticeable differences in the sorting of waste between individual centers in Slovenia. The analysis graphically presents the results of the waste management centers with respect to each attribute or secondary raw material separately (Figure 7).

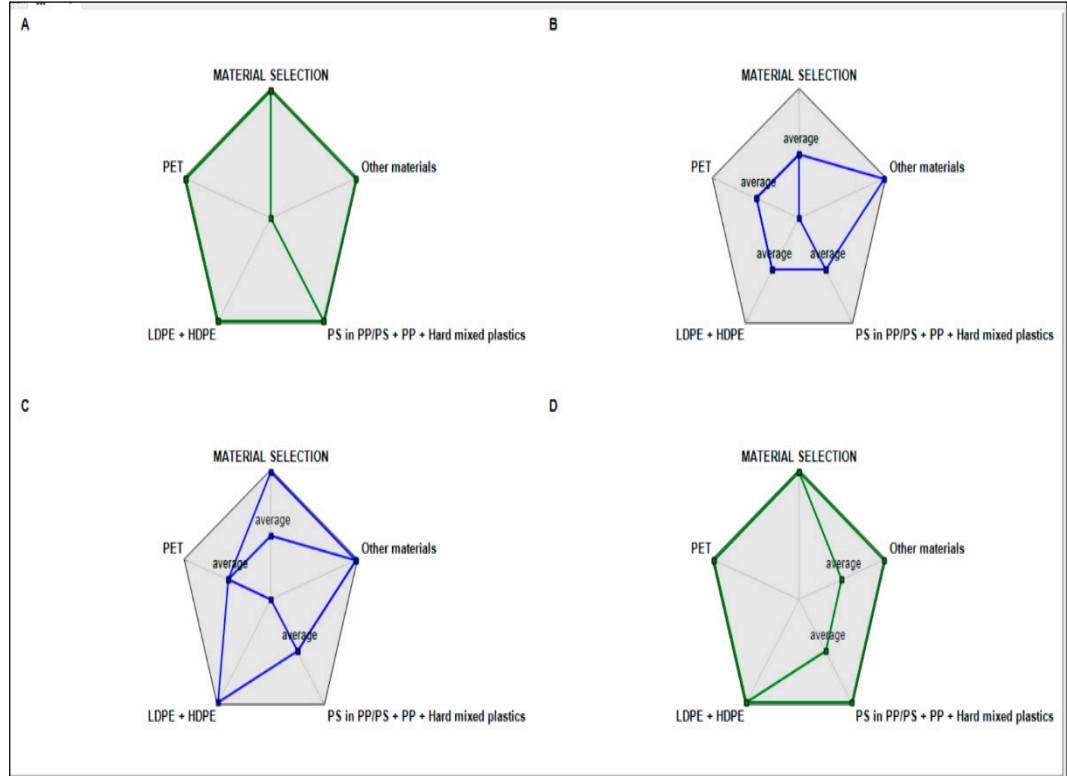

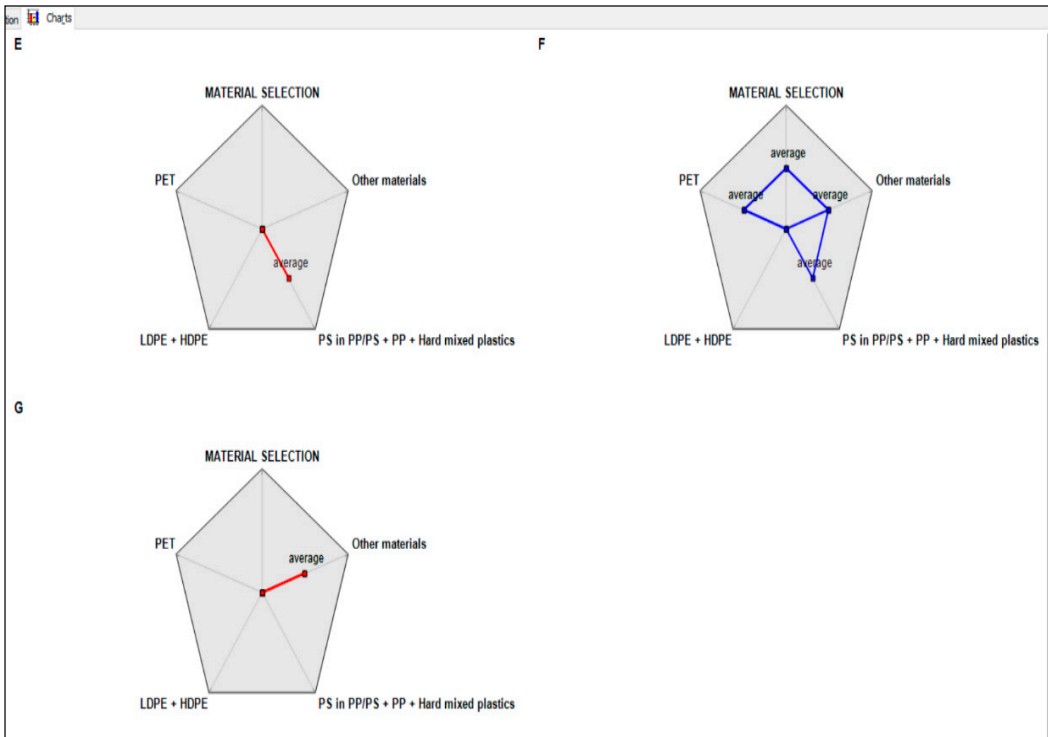

**Figure 7.** Graphical presentation of the first four analyzed waste management centers (from (**A**) to (**D**)). Graphical presentation of the analyzed waste management centers (from (**E**) to (**G**)).

An important feature of using multiattribute decision methodology is the option for detailed analysis of the tree structure of the developed model, analysis of the input data, and analysis of their contribution to overall assessment.

This analysis can also be presented visually with the use of different types of charts. As an example, Figure 7 presents radar charts that show the evaluation of the aggregate attributes of the waste management centers according to the defined decision rules (some of these are shown in Figures 4 and 5). The border of the radar charts represents the best value of the corresponding attribute, while the center of these geometric bodies represents the worst values. The model results closer to the end of the geometric body are better, and vice versa. The results showed that the differences between companies and waste management centers are not minimal and that two of them (E and G) are rated with a poorer rating than "average" (Figure 7).

The ideal waste management center assessment is achieved when the line is at the edge of the pentagram (D). When the line is shifted towards the center, the assessment is non-ideal, and the attribute contributing to non-ideal assessment is clearly visible. This kind of analysis provides important information to decision-makers and enables them to improve the management of waste centers. Furthermore, the model enables the identification of critical points in the process and can therefore play an important role in making necessary changes for improvement.

The results clearly show that the applied multicriteria decision support system approach with DEXi may be effectively applied within the process of ranking and the final quality ranking of municipal waste sorting services in Slovenia.

## 4. Conclusions

Many centers that have been set up using EU funds are responsible for waste management, and these centers have aimed to ensure that at least 50% of all household waste is reused and recycled by 2020.

The research revealed that there are visible differences between analyzed waste management centers. Only two waste management centers were assessed as "good", and there is room for improvement with respect to the reuse of household waste in most cases. MCA aids in making necessary decisions to improve waste management, according to environmental and sustainability goals.

The model used in the case study showed the current shortcomings of certain waste management centers and their advantages. Thus, centers can try to solve their shortcomings, and improve their advantages even further, in order to improve the current situation.

Although some deficiencies of the applied model can be observed (qualitative data only), the approach met our expectations and can be used as a decision support system when making waste management decisions (for instance, when investing in new processing technologies).

The multicriteria decision model described in this paper cannot replace actual decision-makers, but can enable the identification of the best alternatives, according to their own defined criteria and goals. Although complex environmental problems require that many issues are considered simultaneously, the model and its results provide a crucial understanding of observed environmental problems and can efficiently support environmental decision-making in government and private sectors. Further research could be made in combination with the AHP and the analytical network process, including examinations of inter-criteria dependencies and feedback as well as a detailed comparison of the proposed methodology to the existing models. The inclusion of other quantitative methods (such as multiattribute utility theory or multiaspect taxonomic estimation) would also enable precise rankings of waste management centers.

**Author Contributions:** Conceptualization, K.P. and Č.R.; methodology, K.P., Č.R. and S.B.; software, K.P. and Č.R.; validation, K.P. and J.P.; formal analysis, B.P., K.P. and S.B.; investigation, S.B. and K.P.; resources, K.P., S.B., B.P., Č.R. and J.P.; data curation, B.P. and J.P.; writing—original draft preparation, K.P. and Č.R.; writing—review and editing, K.P. and Č.R.; visualization, K.P. and Č.R.; supervision, K.P. All authors have read and agreed to the published version of the manuscript.

**Funding:** This research received no external funding.

**Conflicts of Interest:** The authors declare no conflicts of interest.

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
