# Peer review of "Multicriteria Assessment of the Quality of Waste Sorting Centers—A Case Study"

_sustainability, doi:10.3390/su12093780_

Round 1

Reviewer 1 Report

Title: It is not clear from the title the where was the multicriteria assessment conducted. It highly recommended to be reviewed.

Line 15-16: “The study assessed the quality of the sorting of municipal waste 15 in seven Slovenian waste management centers” this sentence should be revised as it is not clear the primary meaning, What is assessed in the study, the services; the facilities; something else?

Line 17-21: “The assessment …centers” It is a very big sentence which turns to be confusing

Line 28-29 “Because the quantity of solid waste is increasing its management has become a serious issue in 28 modern industrial society”, “because” it is not the most preferred conjunction to use in the first sentence of the introduction, this sentence needs rewriting. Moreover, there are various references that can be used here.

Line 33: “the lack of appropriate environmental assessment tools” an example should be added here by the use of such as… and explain what kind of environmental assessment tools are implied here.

Line 35: “sector (the use of nonrenewable resources and this impacts must be assessed in order to ensure” it seems the ) symbol was forgotten here to close the parenthesis.

Line 35-38: this paragraph as well as the total of the manuscript need academic proofreading. Articles are missing and the terminology used is not sufficient to provide clear meaning.

Line 42-44: “[143 provided a scenario analysis…costs” not clear

Line 47-48, 54-55, 69, 198-200 : reference is missing

Line 58, 66, 79, 83, 96. 104, 108, 109, 114, 117, 119, 157, 250: It would be advisable to use the researchers’ name along with the numbering

Line 71-72: “For efficient solid waste management detailed screening of needs and desired development directions followed by the implementation decision are required” this sentence does not give a clear meaning , “As a result of this such process” necessary for the total of the text to be proof read and reviewed for English language by a native speaker

Line 104-106: “The model will predict an increase in the number of indicators, if a selected number of indicators are not sufficient to distinguish between scenarios and new criterion for the selection of indicators: the relevance of the indicator for certain waste treatment” very big and confusing sentence

Line 111: “A decision-making model must should enable evaluation” in appropriate double use of modal verbs

Line 122: wrong use of the word “concentrate”

Line 128: omit “d”

Line 134:  this phrase must be written in an academic style

Line 136: there is a lack of the verb in the phase “or in the process of closure

Line 139: “or 1,3 kg/ day 7 inhabitant”  better 1.3 , day 7 does not give a clear meaning

Line 141-142: “ i.e 7 kg / inhabitant,and the highest value was found in 2015 (208 kg / inhabitant).” The quantities depict the annual rate or what?

Line 145: is taken over for what?

Line 150: “The appropriate methodology that combined the available data from waste

centers and multicriteria decision about holistic waste sorting management is desirable” is desirable for what? Please specify

Line 186-187: 186 “However, the hierarchical structure of attributes tree for observed problem will help to assessed the quality of the sorting of municipal waste in seven waste management centres” this sentence has major grammar and syntax errors

Line 202: “The value scales by Market” M should be written in lower case

Line 224-225: “In that case some kind of qualitative scale must be defined which is used for determination of qualitative values of each attribute.” Please re-write this sentence there is no clear meaning

Line 233: There is a reference in Figure 5 prior to Figure 4 which is not comprehensible why it was chosen by the author to represent them like this

Line 253: “are more detailed presented” better are presented in more detail

Line 256: “Further” better furthermore,

Line 262-268 please use grammar and syntax rules

Line 282: omit “the

Line 283: meet the requirements should replace the phrase “response their conditions”

Line 293-296: the paragraph needs proof reading

Line 319: a short reference on the relative Directive input would be important here

Line 325, 328: the legends of the two figures have to be in the same content

Line 333: please use the grammar rules

Line 359-360: please specify in which specific cases and under which conditions, the applied model can be as decision support system when making waste management decisions

Line 360: “is intended cannot” please use the proper grammar – syntax type

Line 361: “but can help them discover” should be written in an academic style

Reviewer 2 Report

  1. In those 7 waste sorting centers, besides materials mentioned in the manuscript any other materials are involved? Such as other plastic materials, other metals (other than ALU).
  2. How do you define the scale of price in Figure 3?
  3. Does the scale have three levels (bad, average, good) or 5 levels (+ >= average and <= average)?
  4. It needs to explain how to obtain the importance value in Figures 4 (31%, 31%, 15%, 23%) and 5 (30%, 20%, 20%, 30%).
  5. It needs to explain how to define the decision rules shown in Figures 4 and 5.
  6. It is easy to understand Figure 7, but hard to understand Figures 8 and 9.

Reviewer 3 Report

The subject of the paper is, in my view, interesting: a simple assessment methodology for comparing a large number of (EU) waste sorting centers, or for showing progress of a group of centers, on their success in recovering secondary raw materials from waste. The authors showed that their methodology can be executed in practice and delivers a score that discriminates between good and bad perfromance of sorting centers. I also believe that the assessment parameters fit the context of the EU waste management framework, and so are well-chosen and relevant in today's context.

This being said, I find the quality of the paper itself low. It is difficult to read the paper, because of the large number of syntax and semantic errors in the English, the chaotic line of presentation and because of the ambiguities in the phrasing. 

From a scientific point of view, I miss a lot in this paper:

  • A clear objective: after reading the entire paper, I guess the original objective was to assess the status of Slovenian waste sorting centers with respect to EU targets, as well as potential for improvement; 
  • A discussion of the literature with respect to methodologies that link to the objective, with a clear analysis of where these methodologies are fit or unfit for the authors' objective;
  • An operational definition of the assessment parameters in a way that other authors could repeat the exercise, along with arguments why these parameters lead to the authors' objective; Please note that the present paper repeats exactly the same unclear formulation of assessment parameters four times, in lines 17-21, 123-127, 194-197, 269-272. Most readers will wonder why only output parameters are chosen, and not a ratio of the output to the input. It is quite well-known that waste from rich citizens contains more valuable materials than waste from poor citizens, and so the region or city from which the waste comes is very important for the output, next to the performance of the sorting itself.
  • A link between the three assessment parameters and the four parameters in Figure 5.
  • A discussion and concusion that explain how the underlying differences in the organization or equipment of the centers lead to different scores: this is perhaps the most interesting part for readers.

Next to the missing scientific framework, it is very annoying for readers to see sentences like line 28 (solid waste is increasing) and line138 (quantity of waste decreases every year). 

Reviewer 4 Report

The paper is interesting but in my opinion, there is a great lack of information to complete the research.

These are my comments:

Page 3; Line 111: I think there is a typo (must should)

Page 3: Case study situation:  Authors should explain how many fractions are separated at source in Slovenia

Page 5: Line 178: Authors should explain what “hard mixed plastic “means. What the mixture contains.

Page 6, Figure 2: Please explain better the figure. For example, are these materials collected separately or not? It is not clear in the text.

Page 8 and 9, Figure 3 and 4: The information in this figure is repetitive and can be explained only in the text.

Section 3. Results and discussion:

  • Some information about the separation process in each sorting centre is needed.
  • Authors must explain where the waste come from. It is also necessary a description or characterization of the waste because depending on the origin of the waste, the quality can vary

Page 13, Line 341: Why E?

Section 4: Conclusions. This section is too brief. Please complete them

Round 2

Reviewer 2 Report

The authors did not answer the previous concerns well as stated below, so I still think that it needs the authors’ major revision before the paper can be considered to publish in Sustainability.

1. In those 7 waste sorting centers, besides materials mentioned in the manuscript, any other materials are involved? Such as other plastic materials, other metals (other than ALU).

Authors’ reply: We added additional explanation of the process in table 1. However, Table 1 is nothing to do with materials mentioned.

2. How do you define the scale of price in Figure 3?

My original concern was that in Lines 236~238, the scale of price is not defined well, which means that in what condition(s) can you assign “bad”, “average”, or “good” to the price?

3. It needs to explain how to obtain the importance value in Figures 4 (31%, 31%, 15%, 23%) and 5 (30%, 20%, 20%, 30%).

Authors’ reply: We added additional explanation is lines 283-293. (…As seen in last figure, Market continuity and Weight criteria are the most important (30 %), the similar weight is determined by Quality material and Price criteria (20 %)….). However, it is not explained at all to me.

4. It needs to explain how to define the decision rules shown in Figures 4 and 5.

Authors’ reply: This is additionally explained in lines 283-293. (…The rules were setup together by model developers and experts from each center using multiple discussions. …The rules were determined by the expert group…) However, can the authors provide the logic of the experts for the rules decided?

5. It is easy to understand Figure 7, but hard to understand Figures 8 and 9.

Although the authors improved Figures 8 and 9, however, if some radial lines are not there, they will be easy to be understood.

Reviewer 3 Report

The paper has improved considerably and a lot of comments have been addressed. My recommendation is to now consider the presentation of text, graphs and tables and bring them upto standard. This should remove ambiguities and lines of text that are not understandable at all. Some more interpretation of the assessment outcome in relation to the process design of the facility would be welcome.

Reviewer 4 Report

Dear authors:

Thank you for accepting my suggestions.

Congrats!

Round 3

Reviewer 2 Report

There is an error in Line 169: "use and some waste for disposal. The"

Author Response

Dear Reviewer,

thank you for the comment. We correct suggested: in line 176 (instead of in 169) after the sentence: After sorting, we get waste that is suitable for recycling, waste that is intended for energy use and some waste for disposal. The .... "The" was deleted.

The paper was submitted to MDPI for English editing and has been edited.

Yours sincerely, Karmen PaĹľek.

Reviewer 3 Report

I agree that the paper is worthy of publication with the observation that the English text will be edited to enhance its clarity before publishing.

Author Response

Dear Reviewer,

The paper was submitted to MDPI for English editing and has been edited, as you suggested. 

Yours sincerely, Karmen PaĹľek.